# The Potential Role of Bio Extra Virgin Olive Oil (BEVOO) in Recovery from HPV 16-Induced Tonsil Cancer: An Exploratory Case Study

**DOI:** 10.3390/healthcare13080944

**Published:** 2025-04-19

**Authors:** Ivan Uher, Eva Bergendyová, Janka Poráčová, Jarmila Bernasovská

**Affiliations:** 1Institute of Physical Education and Sport, Pavol Jozef Šafárik University, 04180 Kosice, Slovakia; 2Clinic of Radiation and Clinical Oncology, FNsP J.A. Reiman, Hollého 14, 08001 Presov, Slovakia; e.bergendyova@fnsppresov.sk; 3Faculty of Humanities and Natural Sciences, University of Prešov, 08001 Presov, Slovakia; janka.poracova@unipo.sk (J.P.); jarmila.bernasovska@unipo.sk (J.B.)

**Keywords:** extra virgin olive oil, oncology, recovery, nutritional support, side effects management

## Abstract

The human papillomavirus type 16 (HPV 16) is a high-risk human papillomavirus strain commonly associated with oropharyngeal cancers, including lymph node involvement. The treatment for HPV 16-related tonsil cancer, commonly involving surgery, radiation, and chemotherapy, presents significant challenges. Complications such as oral mucositis, xerostomia, dysphagia, dysgeusia, hypogeusia, impaired gustatory function, and significant weight loss frequently arise, leading to reduced nutritional intake, impaired healing, and recovery progression. These challenges underscore the need for supportive interventions to enhance rehabilitation and the post-recovery period, improve treatment tolerance, and maintain quality of life. **Objective:** This single-subject study examines a 67-year-old male patient diagnosed with a T1N3b (small primary tumor with advanced lump node involvement) associated with HPV 16 positivity, indicating a virus-associated oncogenesis. **Methods:** The patient underwent radiation therapy and chemotherapy, leading to treatment-associated side effects. After having dietary drinks for daily nourishment, the patient routinely incorporated oral bio extra virgin olive oil (BEVOO) to cope with indicated challenges. **Results:** Body composition and metabolic parameters showed treatment-induced declines, followed by substantial but not complete recovery during follow-up examination. Visual Analog Scale (VAS) scores reflected gradual improvements in dysphagia, xerostomia, mucositis, and subtle but ongoing enhancement of the dysgeusia, gustatory perception, and oral palatability. The BEVOO supplementation and mindfulness were associated with positive recovery trends. Additional variables could have impacted the outcomes, preceding and throughout treatment, including the patient’s cognitive and somatic health, environmental conditions, dietary habits, individual attitudes toward recovery, physical activity, and patient way of life. **Conclusions:** These results emphasize the need for additional research employing a comprehensive, multi-factorial framework that accounts for the complex interplay of physiological, psycho-social, and environmental contributors. More extensive, more diverse studies are essential to confirm these observations and substantiate the role of BEVOO as a supportive intervention in cancer recovery.

## 1. Introduction

Human papillomavirus (HPV) infections are widespread, with estimates suggesting that nearly 80% of sexually active individuals will contract HPV at some point in their lives. According to the World Health Organization [1], HPV is responsible for approximately 620,000 cancer cases in women and 70,000 in men [1]. Human papillomavirus (HPV) is a prevalent sexually transmitted infection affecting both men and women. In Europe, the prevalence of genital HPV infection among men varies based on population group. A meta-analysis encompassing studies from Northern, Southern, and Western Europe reported a pooled HPV prevalence of 12.4% among 1863 men in the general population [1]. Among 6428 men considered to be at high risk (e.g., those attending sexually transmitted infection clinics), the pooled HPV prevalence was 30.9%. These findings underscore the significant presence of HPV among men in Europe, with higher prevalence observed in high-risk groups. HPV infection in men can lead to various health issues, including genital warts and cancers of the anus, penis, and oropharynx [2].

Human papillomavirus HPV 16 is a significant oncogenic strain linked to developing oropharyngeal malignancies, particularly tonsil cancer [3]. The incidence of HPV 16-associated tonsil cancer has risen notably in recent years [4], presenting distinct challenges during treatment. Complications such as severe oral inflammation, swallowing difficulties, dry mouth, metallic taste, and substantial weight loss frequently arise [5], which can impair nutritional status, delay recovery, and negatively influence treatment efficacy.

Nutritional interventions are critical in mitigating caloric deficiencies, supporting healing, and maintaining body weight [3]. Research in this area has extensively explored using high-calorie liquid supplements [6], protein-rich diets, and other dietary modifications [7] to support cancer patients through their course of treatment. BEVOO is a key component of the Mediterranean diet and offers antioxidant, anti-inflammatory, and anticancer properties [8] that can benefit oncology patients. Rich in monounsaturated fatty acids and polyphenols, it helps combat oxidative stress and inflammation, contributing to cancer progression and treatment side effects. Studies suggest [9,10,11] that regular olive oil consumption may lower the risk of breast, colorectal, and prostate cancer by promoting apoptosis, reducing DNA damage, and inhibiting tumor proliferation. Additionally, its immune-modulating and gastrointestinal health benefits [12] can support oncology patients undergoing chemotherapy or radiation therapy by reducing side effects like mucositis, cachexia, and metabolic imbalances [13]. The existing body of evidence on olive oil focuses primarily on its role in cardiovascular health and chronic disease prevention [14]. The beneficial properties of certain food compounds, especially BEVOO, in cancer recovery, and its potential application as a supportive nutritional oncology intervention remain unexplored [15]. Specifically, its ability to address complications like xerostomia, dysphagia, dysgeusia, hypogeusia, and mucositis while improving caloric intake and weight management warrants further investigation. Moreover, a study by [16,17,18,19] concluded that nutritional support (omega-three fatty acids) plays a vital role in Allogeneic Hematopoietic Stem Cell Transplantation (allo-HSCT), although optimal management has not yet been well established. The presented exploratory case study aims to investigate the use of BEVOO as an adjunct to nutritional intervention in a patient undergoing treatment for HPV 16-related tonsil cancer. By examining its potential benefits and role in supporting recovery, this study seeks to contribute to the emerging field of personalized nutritional strategies in cancer care.

## 2. Methodology

This single case study was conducted over 6 months in a home setting with regular monitoring of participant responses by medical professionals at the hospital to evaluate the potential supportive role of BEVOO in the treatment and recovery of a patient with HPV-related tonsil cancer. In January 2024, a 67-year-old male observed a small lump on the left side of the neck, located in the lymph node region. Following the progressive enlargement of the lesion, in May 2024, a CT scan detected an HPV 16-associated malignant neoplasm of the lymph node (T1N3b), classified as a T1 early-stage carcinoma with metastasis to the left lymph node. Comprehensive radiation and chemotherapy were performed between 3 July and 20 August 2024. On 28 June 2024, laboratory results indicated an elevated serum glucose (S-GLU) level of 6.19 mmol/L, with other clinical values within normal limits. The patient had no history of smoking, alcohol consumption, or illicit drug use. He had followed a vegetarian diet for four decades and was physically active. He also had no record of contact with environmental contaminants or chronic stress-inducing factors. Before initiating treatment, the patient was not on any regular medication, except for intermittent use of ibuprofen 400 mg, a dosage of two pills daily a month prior to undergoing radiation and chemotherapy to manage pain caused by the neck tumor—no known allergies. The mother succumbed to cholangiocarcinoma in her eighties. The father passed away from heart failure in his mid-80s.

The patient underwent concurrent radiation and chemotherapy as the primary treatment. Radiation treatment involved external beam radiation at a total dose of 70 Gray (Gy) over 35 sessions. The chemotherapy regimen included Cisplatin (cDDP) 100 mg/m^2^ administered IV every 21 days × 2 cycles. The treatment period spanned from 3 July to 20 August 2024. Due to the advanced effects of radiation and chemotherapy, including a persistent metallic taste and reduced appetite, the patient’s food intake was limited to regular consumption of three to four bottles of Nutra drink Resource^®^ 2.0 Fiber per day. Each 200 mL serving provides 400 kcal. To support caloric intake and provide essential nutrients, the patient consumed 30 to 45 mL, 240–360 kcal of BEVOO (i.e., 2–3 tablespoons daily), administered post-liquid meals, starting one week before therapy, and continuing throughout the treatment and recovery period. The bio extra virgin olive oil used in this study was produced in Germany from organically grown olives, using first cold-pressing mechanical extraction methods. The oil is certified organic according to IT-BIO-006 EU Landwirtschaft. It is composed primarily of monounsaturated fatty acids, especially oleic acid (70–80%), along with polyunsaturated fatty acids such as linoleic acid (10%) and saturated fats like palmitic acid (10–15%). It also contains minor but biologically active components including polyphenols, tocopherols, and squalene. The BEVOO used in this study was selected based on the literature and producer information indicating a high polyphenol content (typically > 250 mg/kg). Although the exact polyphenol concentration of the batch used was not independently analyzed, the product is known for its consistently high antioxidant profile, as reported in previous studies. Compliance with BEVOO supplementation was tracked using daily patient logs. Side effects of the primary treatment were managed with supportive care (laxatives—the need for laxatives arose on four occasions throughout the treatment; baking soda—rinses to maintain oral hygiene; and regular oral hydration and Fumukal two times daily to moisturize, lubricate, and cleanse the oral cavity, contributing to symptom relief and improved oral hygiene during treatment). No adverse effects related to BEVOO were observed. The intervention was monitored, with adjustments planned based on the patient’s evolving tolerance and nutritional status. The visual timeline is presented in Table 1. Data were collected using both subjective and objective measures. The patient’s psychological state was assessed at baseline using multiple assessment tools. Multidimensional assessment of interoceptive awareness (MAIA Version2) [8] was used to evaluate the patient’s interoceptive awareness across eight subscales. The Hospital Anxiety and Depression Scale (HADS—A/D) [9] focused on both the anxiety and depression subscale. Moreover, the Interoceptive and Cognitive Experience Questionnaire (ICE-Q) assessed the subject’s tendencies to overthink, ability to live in the present moment, and awareness of internal bodily sensations was facilitated. Two of the questionnaires were administered via the internet using an online platform, while the third ICE-Q was conducted in a traditional paper-based format. To ensure the results of the administered questionnaires were meaningful and accurate, researcher provided clear instructions to the patient and ensured he understood how to respond (choose the option that best reflects how he has felt in the past week). We employed the same environment and conditions for administration to minimize external influences. At the start of the treatment, the questionnaires were administered twice at a one-week interval to evaluate test–retest reliability, where minor variability was observed and deemed meaningful. The patient completed the questionnaire assessment using a web-based application under the supervision of a researcher. Body weight and composition were monitored daily under standardized conditions, with weekly hospital-based measurements as a validation reference. While consumer-grade analyzers may have minor errors, the focus was on tracking trends rather than absolute values, which remains a reliable approach in longitudinal assessment. Given the consistency in data collection and the alignment with hospital-based measurements, we are confident that the recorded data accurately reflects the participant’s physiological variables.

Records were synchronized to a mobile device via the Vital Body Plus application for subsequent analysis. A scale provided detailed metrics such as measuring body weight, water percentage, visceral fat, subcutaneous fat, muscle mass, BMI, muscle index, protein ratio, and basal metabolic rate (BMR). Equipped with advanced sensors and analytical capabilities, it is suitable for monitoring changes in body composition over time, making it suitable for clinical and personal use. Body composition parameters and metabolic indicators were measured at three key time points: at the start of treatment (baseline), at the end of treatment, and during the follow-up recovery period (week 12). Daily measurements were initially collected using the Vital Body Plus application to capture short-term variations. However, due to the gradual changes observed during this study, intermediate daily and weekly values were averaged and summarized into these three points to provide a concise overview of the trends. The patient reported symptoms of xerostomia, dysphagia, and mucositis that were assessed weekly using a Visual Analog Scale (VAS). The patient was instructed to rate the severity of each symptom on a 10-point linear scale, with 0 representing no symptoms and 10 representing the worst imaginable severity. The VAS was chosen for its simplicity, sensitivity, and reliability in tracking symptom changes over time.

The data were collected on a daily basis throughout the study period. For the evaluation of outcomes, basic descriptive statistics were used frequencies and percentages. These were employed to summarize the data and observe general trends over time. No advanced statistical testing was applied, as the focus was on monitoring and reporting patterns in the collected data.

The participant provided informed consent prior to participation. The consent included an acknowledgement of this study’s purpose, procedure, potential risks, and benefits. The researcher was fully aware of his right to discontinue this study at any time. Ethical approval for this case study was granted by the ethical committee of Prešov University for Research. ID: ECUP102024PO on 19 December 2024. This study was conducted in accordance with the ethical principles outlined in the Declaration of Helsinki. The reporting of this case study adheres to the CARE guidelines as recommended by the EQUATOR Network. The patient tracked the body composition and metabolic parameters daily at home using a multifunctional digital scale, the Vital Trainer Analytical Scale ETA 778090000.

## 3. Results

### 3.1. Anthropometric Characteristics

The recorded parameters indicated associated changes over the course of radiation and chemotherapy treatment (Table 2), reflecting the impact on the patient’s overall nutritional and metabolic status. At the onset of radiation and chemotherapy, the patient’s baseline weight was 72 kilograms (kg), with a body mass index (BMI) of 21.3 kg/m^2^ indicating a standard nutritional status without signs of malnutrition. During the course of treatment, dysgeusia, mucositis, xerostomia, dysphagia hypogeusia, altered taste modulation, and impaired gustatory perception contributed to a decline in oral food intake. By the end of the treatment, the patient’s weight had decreased to 63 kg, representing a total weight loss of 12.5%. This degree of unintentional weight loss was clinically significant and indicative of a heightened risk of malnutrition. Thus, factors contributing to this weight loss included both reduced caloric intake due to treatment side effects and the potential for increased metabolic demands associated with cancer and its therapy. The decline in nutritional status underscored the necessity of timely nutritional intervention to mitigate further weight loss and support recovery, especially in patients with moderate physiques before undergoing treatment.

At the start of treatment, the patient’s body water percentage was 62.2%, which increased to 65% by the end of the treatment period. This 4.5% increase indicates fluid retention, a common side effect of treatment-related inflammation or reduced protein intake potentially affecting water distribution within the body. The patient’s visceral fat level decreased from 7% at baseline to 5% post-treatment, highlighting a reduction in fat stores around internal organs of 28.7%. Subcutaneous fat levels also significantly declined, from 14.4% to 10.4%, indicating an overall loss of 27.78% of fat reserves. Muscle mass decreased from 31.3 kg at baseline to 29.5 kg post-treatment, reflecting a 5.8% reduction. This loss may be attributed to reduced protein intake, increased metabolic demands, and the catabolic effects of cancer treatment. The patient’s BMR dropped 7.69% from 1508 kcal/day to 1392 kcal/day, consistent with reduced muscle mass (sarcopenia) and overall body weight. The downturn stresses the urgency of implementing calorie and protein-focused dietary support to mitigate metabolic deceleration and muscle wasting. Despite significant weight loss through radiation and chemotherapy, the patient’s muscle index increased from 43.1 to 46.8 units, and the protein ratio rose 8.58%. The protein ratio increased from 18 to 18.5 during treatment, representing a 5% enhancement. Even with the physiological challenges brought on by treatment, this increase could stem from the effective role of BEVOO and metabolic shifts that assist the body’s heightened need for healing and regeneration. Preserving or enhancing protein levels during treatment is essential for preventing muscle loss, bolstering immune response, and facilitating tissue recovery. These findings highlight the significance of personalized nutritional strategies to maximize patient outcomes during rigorous therapy. Measurements between the end of treatment and the follow-up evaluation demonstrate substantial shifts in physiological body measurements. Body weight increased from 63 kg to 69 kg, indicating a 9.52% improvement. Body water showed a slight decrease from 65% to 63%. Visceral fat increased from 5% to 6%, while subcutaneous rose from 10.4% to 13.3%. Muscle mass slightly increased from 29.5 kg to 30.7 kg, reflecting a 4.07% improvement. BMR increased significantly, from 1392 kcal to 1472 kcal. The BMI rose from 18.6 to 20.5, reaching a healthy range. However, the muscle index decreased from 46.8 to 44.2, and the protein ratio declined slightly from 18.9% to 18.3%. These data highlight a range of tendencies, from weight gain, better BMR, and muscle mass improvements to minor reductions in body water, muscle index, and protein ratio, that may merit closer evaluation. A consecutive laboratory test on 9 August 2024 revealed an improvement in S-GLU, which decreased to 5.65 mmol/L, compared to the initial value of 6.19 mmol/L recorded on 28 June 2024. The decrease in S-GLU suggests an improvement in glucose regulation, possibly due to dietary changes and metabolic adjustments. However, there was an elevation in serum urea (S-UREA) to 13.60 mmol/L. The increase in S-UREA may indicate altered kidney function, dehydration, increased protein metabolism, or dietary protein intake, and further evaluation may be needed to determine the cause, especially if the elevation persists. Since other values remained within normal limits, the change may be isolated and influenced by physiological factors rather than a systemic pathology. The specific intervention, incorporating BEVOO and liquid nutrition, may have played a role in maintaining lean body mass.

Furthermore, alterations in body water balance, leading to higher intracellular water retention, might explain the observed variations in muscle index and protein ratio. This highlights the potential for nutritional strategies to mitigate excessive muscle depletion and maintain protein levels during oncological treatment.

### 3.2. Oral and Swallowing-Related Symptoms

The severity of xerostomia, dysphagia, and mucositis was monitored weekly using the Visual Analog Scale (VAS) over the seven weeks of treatment (Figure 1). The collected data reveal distinct trends in symptom progression and resolution. Xerostomia exhibited a steady but gradual improvement throughout the treatment period. The VAS scores began at 6.5 in week 1, reflecting significant dry oral cavity symptoms, and progressively decreased to 5.0 by week 7. This indicates that while xerostomia remained a persistent issue during the treatment, the severity lessened over time, suggesting some degree of adaptation or response to interventions, such as BEVOO. Dysphagia followed a similar improving trend but recovered at a faster rate compared to xerostomia. The initial VAS score of 6.8 in week 1 indicates difficulty swallowing, likely exacerbated by mucosal irritation from radiation therapy. By week 7, the score had notably advanced to 3.5, demonstrating a moderate to significant reduction in swallowing difficulty. This suggests the efficacy of supportive measures and recovery mechanisms in addressing inflammation and tissue damage affecting swallowing. Furthermore, mucositis was most severe during the initial stages of treatment, with a VAS score of 7.0 in week 1. The severity decreased more rapidly compared to the other symptoms, with scores dropping to 4.5 by week 4 and reaching a low of 1.5 by week 7. The rapid resolution of mucositis suggests effective inflammation and tissue healing management, potentially accelerated by the anti-inflammatory and antioxidant properties of BEVOO.

Moreover, during the follow-up assessment period (weeks 8–12) after the completion of radiation and chemotherapy, a continued decline in mucositis, dysphagia, and xerostomia was observed. By week 12, the mucositis severity level was 0.5, dysphagia decreased steadily to 2.3, and xerostomia dropped to 4.0. This extended decline likely reflects the gradual recovery of damaged tissues, acute inflammation reduction, and resolution of treatment-related side effects as the body regains homeostasis.

Though at varying rates, all three symptoms consistently improved over the treatment period. Mucositis resolved the fastest, with significant reductions observed as early as week 3. That aligns with the typical healing trajectory of mucosal tissues following radiation therapy cessation. Dysphagia showed intermediate improvement, reflecting gradual relief from inflammation and improved functionality in swallowing mechanisms. Xerostomia exhibited the slowest recovery, with persistent symptoms evident throughout the treatment period. That is consistent with the delayed regeneration capacity of salivary glands following radiation damage. The improvement in mucositis after the treatments (8–12 weeks) can be attributed to the cessation of cytotoxic treatments, allowing for epithelial regeneration and the resolution of mucosal damage. Similarly, the decline in dysphagia severity may indicate recovery of oropharyngeal structures and the re-establishment of standard swallowing mechanics. The alleviation of xerostomia symptoms is likely attributed to the partial recovery of salivary gland function or adaptive compensatory processes over time. These findings emphasize the significance of follow-up assessments in observing the course of symptom recovery post-therapy. Multiple studies [20,21] suggest that BEVOO compounds demonstrate potent antioxidant activity and significantly contribute to the prevention and suppression of chronic degenerative diseases linked to inflammatory pathways, including cardiovascular, brain, and cancer diseases. The synergy of these factors may partially account for the observed improvements in patient well-being after undergoing radiation, chemotherapy, and recovery period.

### 3.3. Findings in Psychological Profiles

Furthermore, when evaluating psychological traits, the patient’s 4.6 rating on the MAIA scale reflects a moderate to substantial interoceptive perception. This measurement implies that the patient could frequently recognizes and deciphers internal bodily cues, applying this awareness to effectively regulate emotional responses and actions. This level of interoceptive awareness plays a crucial role in supporting mindfulness, coping with treatment-related challenges, and facilitating recovery. They scored four on the HADS-A (Anxiety) subscale and three on the HADS-D (Depression) subscale, both indicating very low levels of anxiety and depression. These scores fall well within the normal range, suggesting the patient is emotionally stable with no signs of significant distress. The results of the ICE-Q questionnaire indicated a reduced tendency toward overthinking, a strong focus on present-moment experiences, and heightened interoceptive awareness. The overall results of these questionnaires indicate a state of mindfulness that likely enhanced the patient’s ability to regulate challenges and make decisions that supported his treatment and recovery period. It ought to be noted that the questionnaires were administered only at the beginning of treatment to establish the patient’s baseline mental state. It was hypothesized that the mental state at the onset would remain relatively stable over a short period of time (i.e., course of treatment) but could significantly influence the patient overall behavior and engagement during treatment. This approach focused on understanding the initial psychological variables of the patient that might impact the treatment process rather than tracking changes over time. Furthermore, this single administration minimized the burden on the patient while providing critical insights for guiding care.

## 4. Discussion

The detected shifts in body composition emphasize the physiological strain caused by chemotherapy and radiation therapy, which is followed by a progressive recovery during the follow-up phase. The significant decrease in body weight (72 kg to 63 kg) and BMI (21.3 to 18.6) during treatment reflects the expected impact of treatment on nutritional status and energy reserves. However, the partial recovery in weight (69 kg) and BMI (20.5) during follow-up indicates a positive trend, likely supported by metabolic recovery. In general, the results underscore the importance of a well-rounded recovery strategy that prioritizes protein consumption, proper hydration, and regular physical activity to aid lean mass regeneration and metabolic efficiency while preventing excessive weight reduction. The progressive decline in mucositis, dysphagia, and xerostomia throughout radiation therapy and chemotherapy, followed by the 8–12 weeks recovery period, illustrate the dynamic interplay between treatment-induced side effects and the body’s capability for healing. These changes reflect the cessation of cytotoxic treatment, allowing for epithelial regeneration, resolution of inflammation, and partial restoration of salivary gland function. Dysphagia improvement can be attributed to structural recovery and the re-establishment of swallowing mechanics. At the same time, the slower decline in xerostomia severity highlights the long-term impact of salivary gland damage and the need for continued supportive care. Nutritional rehabilitation may play a vital role in supporting and promoting recovery, and improving the quality of life in cancer patients, particularly those experiencing treatment-induced complications such as mucositis, xerostomia, and dysphagia. Effective nutritional care helps prevent correct malnutrition, supports immune function, and enhances the body’s ability to tolerate and respond to therapy [22,23]. Early identification of nutritional risk and individualized interventions, ideally implemented by trained professionals such as dietitians and clinical nutritionists, are key to successful rehabilitation. Despite its importance, nutritional care remains insufficiently integrated into routine oncological management [24,25] in many healthcare systems, including Slovakia, indicating a need for structural changes and greater emphasis on multidisciplinary collaboration. Furthermore, comprehensive cancer care requires an integrated strategy involving medical doctors, nutritionists, dietitians, lifestyle specialists, and psychologists to address patients’ complex physical and psychological needs. This integrated effort is essential for optimizing nutritional status, managing treatment-related side effects, and supporting overall well-being. Nonetheless, these multidisciplinary teams in the Slovak healthcare system are often limited or absent, particularly in routine oncology care. Strengthening this aspect of care represents a critical opportunity for improving patient outcomes and quality of life.

It is to be emphasized that the use of BEVOO appeared to mitigate one of the common side effects of radiation and chemotherapy, a metallic taste that often deters patients from consuming solid foods. By providing a neutral, palatable alternative, BEVOO enables the patient to maintain some level of oral intake without exacerbating an aversion to food. This aligns with existing research [26,27,28] suggesting that BEVOO’s mild flavor and bioactive properties may help neutralize altered taste sensations caused by treatment-related mucosal changes or dysgeusia.

Moreover, the inclusion of BEVOO in our single-subject study potentially prevented rapid weight decline, a critical factor for maintaining ideal body weight and energy balance and ensuring adequate treatment adherence. Although the patient experienced difficulty consuming solid foods, including Nutra drink Resource ^®^ 2.0 Fiber, which was not completely well-tolerated, it was better tolerated than solid food that the patient did not consume. BEVOO was provided as a source of calories and essential nutrients. It was moderately easily incorporated into direct consumption, allowing the patient to avoid complete nutritional deprivation during intense treatment side effects and recovery time.

The patient’s acceptance of BEVOO highlights its potential as a supportive dietary intervention. Unlike other nutrient-dense options that were challenging to consume due to altered taste or mucosal discomfort, BEVOO provided a non-bothersome, more tolerable alternative. This suggests that BEVOO may be a viable option for patients experiencing oral mucositis, dysgeusia, or metallic taste, which often limit the dietary variety and contribute to malnutrition. This is for patients with a normal body weight, especially those closer to the lower end of the healthy weight spectrum. BEVOO use effectively reduces the risk of significant weight loss, which is more pronounced in these individuals than in those with higher body mass.

In sum, the ability of BEVOO to mitigate the metallic taste and its palatability may have indirectly contributed to the stabilization and slowed down the retardation of nutritional status and weight maintenance. This outcome is particularly significant in the context of cancer care, where rapid weight loss and malnutrition are associated with poorer treatment outcomes and increased morbidity [22,29,30]. The caloric contribution of BEVOO (30–45 mL, approximately 240–360 kcal from 2–3 tablespoons daily) provided an accessible energy source without increasing the patient’s discomfort, enabling better management of treatment-related side effects. These findings highlight the importance of identifying patient-specific, tolerable dietary options during cancer therapy. BEVOO’s unique properties, both as a palatable, nutrient-dense food and as a source of bioactive compounds with potential anti-inflammatory and antioxidant effects, position it as a promising adjunct to standard nutritional interventions in oncology [31].

In addition to the potential physiological effects of BEVOO on treatment and recovery, by maintaining a balanced mental state through interoceptive awareness and living more in the present moment, our research subject could in some way contribute to the patient’s ability to cope with treatment challenges [32,33,34]. It can be argued that mindfulness potentially enhanced awareness of internal signals, helped the patient capture reality more accurately, reduced emotional distress, and facilitated a sense of control. Consequently, it helps better equip the patient to make informed decisions, support them, and facilitate a more efficient recovery process. It can be concluded that the incorporation of BEVOO not only provided physiological benefits, but also enhanced physical awareness that supported recovery. Altogether, the patient subjectively reported that the consumption of BEVOO was the most tolerable nutritional component during treatment. Findings from study [35] suggest that BEVOO may represent a beneficial approach for cancer patient care. Another investigation [36] revealed that a sustainable dose of n-3 FAs (EPA-DHA) at or above 1.5 g/day is beneficial for enhancing markers and overall quality of life in those with cancer. Study [37] highlights the development of fat-rich nutritional formulas as a promising advancement in cancer patient support. The patient also perceived BEVOO as contributing to a sufficient daily nutritional load, which may indicate a placebo effect or a psychological reinforcement of dietary adequacy. Hence, this perceived tolerability and nutritional sufficiency contributed to enhanced autonomy and control over dietary management, potentially improving adherence to nutritional recommendations during treatment [38,39,40]. While outlined clinical observation provides preliminary insights into the potential benefits of BEVOO during treatment and recovery from HPV 16-related tonsil cancer, these findings are exploratory and require further validation.

The strength of the presented study can include a novel approach to nutritional support in the treatment and recovery from tonsil cancer. Multisystem BEVOO’s impact on skeletal and GI systems provides a holistic view of its benefits. The findings have practical implications for shaping dietary interventions and addressing potential psychological and behavioral benefits that are often overlooked in nutritional studies. This study bridges the gap between basic science and clinical application–patient-reported outcomes and recovery strategies. Limitations include the potential for placebo or other dietary fats to differentiate its unique effects, subjective bias, limited mechanistic data (e.g., biomarkers of inflammation, gut microbiota composition changes), sample size, and generalizability. Differences in genetic, metabolic, and disease-related factors may influence how patients respond to BEVOO; this key limitation should be addressed in future research. The overall impact of BEVOO’s effects on different physiological systems is provided in (Table 3).

## 5. Conclusions

The presented investigation highlights the profound impact of chemotherapy and radiation therapy on body composition, metabolic parameters, and overall health of a patient with an ICD-10 diagnosis of C09.8. It also underscores the potential role of supporting interventions such as BEVOO supplementation on muscle preservation, metabolic health, inflammation control, resilience, and adaptability in challenging situations. The patient’s body composition and metabolic parameters demonstrated significant treatment-induced changes, including reductions in weight, muscle mass, and basal metabolic rate, alongside shifts in fat distribution and protein ratio. During the follow-up period, partial recovery was observed, with body weight, BMI, muscle mass, and metabolic activity improvements. However, persistent imbalances in muscle index and fat composition suggest a need for tailored interventions to optimize lean body mass restoration and metabolic stability. VAS scores and body composition data provided a valuable perspective on patient-reported outcomes and recovery dynamics. The gradual changes observed emphasize the importance of consistent monitoring and holistic approaches to recovery. BEVOO emerged as a potential beneficial factor, with its multi-beneficial properties contributing to improved body composition and energy balance during treatment and recovery. Additionally, the perception of BEVOO as a tolerated and beneficial dietary component may enhance control over nutrition, contributing to improved psychological resilience and adherence to dietary regimens during treatment and recovery. These findings suggest that integrating nutritional interventions like BEVOO and a proper mindset can create a synergistic effect that enhances both physical and mental recovery. BEVOO shows promise for inclusion in treatment and recovery protocols for patients experiencing sarcopenia, cachexia, radiotherapy and chemotherapy-induced side effects, metabolic dysfunction, and neuroinflammation. Strategies that combine metabolic and body composition monitoring with personalized interventions offer a pathway to more compelling, comprehensive recovery plans for patients.

## Figures and Tables

**Figure 1 healthcare-13-00944-f001:**
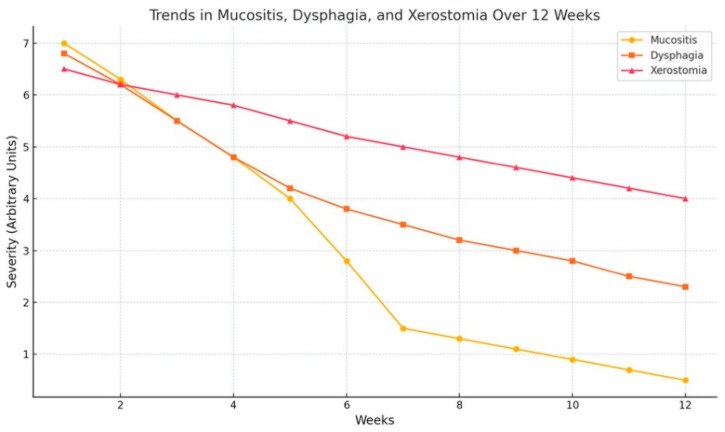
Improvement in Treatment-Related Symptoms (VAS Scores).

**Table 1 healthcare-13-00944-t001:** Visual timeline.

Time Period	Event/Activity
Week 1	BEVOO supplementation started, baseline questionnaires.
Weeks 1–7	Radiation and chemotherapy administration, daily monitoring.
Week 8	End of the treatment phase, BEVOO continued.
Weeks 9–12	Follow-up monitoring and reassessment.

**Table 2 healthcare-13-00944-t002:** Changes in Body Composition and Metabolic Parameters During Radiation and Chemotherapy and Follow-up Assessment.

Parameters	Baseline (Start of Treatment) 3 July 2024	End of Treatment 20 August 2024	Change	Follow-Up Evaluation 17 September 2024	Change
Weight (kg)	72	63	−9 kg (−12.5%)	69	+6 kg (9.52%)
Body Water (%)	62.2	65	+2.8 (+4.50%)	63	−2 (−3.08%)
Visceral Fat (%)	7	5	−2 (−28.7%)	6	+1 (+20%)
Subcutaneous Fat (%)	14.4	10.4	−4 (−27.78%)	13.3	+2.9 (+27.88%)
Muscle Mass (kg)	31.3	29.5	−1.8 kg (−5.80%)	30.7	+1.2 kg (+4.07%)
BMR (kcal/day)	1508	1392	−116 kcal (−7.69%)	1472 kcal	+80 (+5.75%)
BMI	21.3	18.6	−2.7 BMI (−12.68)	20.5	+1.9 (+10.22%)
Muscle Index	43.1	46.8	+3.7 (+8.58%)	44.2	−2.6 (−5.56%)
Ratio of Proteins	18	18.9	+0.9 (+5.00%)	18.3	−0.6 (−3.17%)

Note. This table was created by the author based on collected data.

**Table 3 healthcare-13-00944-t003:** Effects of BEVOO on bodily systems.

System/Process	Effect	Mechanism/Notes	Refs. No.
Anticancer and Chemopreventive Properties	- Reduces cancer cell proliferation and promotes apoptosis. - Exhibits chemopreventive effects in various cancers (e.g., breast, colon, and prostate).	- Contains oleocanthal, which induces cancer cell death without harming healthy cells. - Polyphenols and squalene act as antioxidants and inhibit tumor growth and metastasis.	[11,41,42]
Antimicrobial and Chemotherapeutical effects	- Act as a natural antimicrobial agent.	- Inhibits the growth of bacteria, fungi, and viruses due to phenolic compounds.	[43,44]
Cardiovascular System	- Improves lipid profile and reduces LDL oxidation. - Supports blood pressure regulation.	- Antioxidant properties reduce oxidative stress; polyphenols enhance vascular health. - Rich in oleic acid, which enhances endothelial function and reduce inflammation.	[45,46,47]
Autonomic Effects	- Modulates autonomic nervous system activity.	- Potential calming effect on sympathetic activity due to its anti-inflammatory and metabolic benefits.	[48,49]
Central Nervous System (CNS)	- Supports cognitive function and reduces neurodegeneration risk. - Exhibits neuroprotective effects in aging and neurodegenerative diseases. Is neuroprotective, analgesic, and antinociceptive. - Induce epigenetics.	- Rich in oleocanthal and other polyphenols, which protect neurons and reduce amyloid-beta plague formation. - Antioxidants combat oxidative stress: anti-inflammatory properties reduce neuroinflammation.	[50,51,52]
Anti-inflammatory Properties	- Reduces systemic and localized inflammation. - Alleviates inflammation in chronic conditions such as arthritis and metabolic syndrome.	- Polyphenols inhibit inflammatory mediators like prostaglandins and cytokines. - Oleic acid and squalene target pathways involved in chronic inflammation.	[53,54]
Endocrine System	- Improves insulin sensitivity and glucose metabolism. - Regulates hormone production and balance.	- Monounsaturated fats and bioactive compounds enhance insulin signaling and pancreatic function. - Antioxidant and anti-inflammatory effects influence endocrine pathways.	[55,56]
Immune System	- Enhances immune response and modulate inflammation. - Exhibits antimicrobial properties against pathogens.	- Anti-inflammatory polyphenols like oleuropein reduce pro-inflammatory cytokines. - Antibacterial compounds, such as hydroxytyrosol, inhibit microbial growth.	[57,58]
Skeletal System	- Enhances calcium absorption. - Preserves bone mineral density. - Mitigates muscle atrophy. - Reduces risk of sarcopenia. - Helps counteract cachexia.	- Increases bioavailability of calcium and supports vitamin D metabolism. - Polyphenols promote osteoblast activity and reduce osteoclast-mediated bone resorption. - Monounsaturated fats and polyphenols protect muscle fibers. - Support muscle protein synthesis, improving metabolic balance and reducing catabolic pathways. - Anti-inflammatory properties help mitigate tissues wasting in cancer-related cachexia.	[59,60]
Gastrointestinal System (GI)	- Modulates gut microbiota. - Improves fat digestion and nutrient absorption.	- Polyphenols enhance beneficial gut bacteria (Bifidobacterium, Lactobacillus), improving gut health. - Stimulates bile secretion, aiding fat digestion and absorption of fat-soluble vitamins (A, D, E, K).	[61,62,63]
Respiratory System	- Reduces airway inflammation and supports lung health.	- Olive phenols anti-inflammatory properties decrease pro-inflammatory cytokines involved in respiratory conditions.	[57,64]
Antioxidant Effects	- Neutralizes free radicals and protects cellular integrity.	- Rich in vitamin E and polyphenols, which scavenge reactive oxygen species and reduce oxidative damage.	[21,65]

Data collected and analyzed by the authors.

## Data Availability

The authors have full access to all specific material used in this paper and take responsibility for the use and accuracy of the information provided.

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
