# Peer review of "The Potential Role of Bio Extra Virgin Olive Oil (BEVOO) in Recovery from HPV 16-Induced Tonsil Cancer: An Exploratory Case Study"

_healthcare, 2025, doi:10.3390/healthcare13080944_

Round 1
Reviewer 1 Report
Comments and Suggestions for Authors
Dear Authors,
Thank you for submitting this manuscript, which explores a very interesting topic. I believe the manuscript aligns well with the aims of the journal, and I provide the following suggestions and comments:
ABSTRACT:
- I suggest shortening the abstract and structuring it into the following sections: Objective/Scope, Methods, Results, and Conclusions.
Keywords:
- Remove "oil supplements." I recommend inserting "Extra Virgin Olive Oil" instead.
INTRODUCTION:
- I suggest expanding this section by emphasizing the epidemiological aspects, particularly by including data on HPV.
- Line 56-57: I recommend expanding this part and including a citation to support the claims.
- Line 59-63: I suggest expanding this section. Olive oil contains omega-3 fatty acids, specifically in the form of linoleic acid, a polyunsaturated fat. These effects, particularly the anti-inflammatory and immunomodulatory actions, have been studied extensively, especially in the context of oncology and hematological diseases. I suggest expanding the introduction to include the effect of omega-3 in patients undergoing bone marrow transplantation.
METHODS:
- I believe this section is adequately described. However, I recommend presenting the study in accordance with the EQUATOR guidelines for case studies.
- Vital Trainer Analytical Scale ETA 778090000: Please describe the use of this tool more clearly. If it is a home-based impedance scale, the data may be subject to bias since it is not a certified medical device.
- Lines 106-120: Please specify the units of measurement for the anthropometric parameters.
RESULTS:
- The authors have described this section clearly; however, I believe it would improve readability if the results section were divided into subsections for better clarity.
- In the results, you mention dysgeusia, a complication that significantly impacts these patients. Why wasn’t this assessed with a specific scale, such as the CITAS scale?
DISCUSSION:
- I suggest reducing the length of this section and emphasizing a discussion of more recent international studies to support your findings.
- One aspect that has not been considered, but would warrant an extensive discussion supported by recent studies, is the topic of nutritional prehabilitation in head and neck cancer.
- Additionally, you could expand the discussion regarding the importance of a multidisciplinary team in supporting cancer treatments and nutritional care in this patient population. The involvement of medical doctors, nutritionists, dietitians, and hospital nutrition teams, as well as specialized roles such as clinical nurse specialists in nutrition, should be emphasized.
Reviewer 2 Report
Comments and Suggestions for Authors
This great case report examines the effect of an olive oil-based food supplement on head & neck cancer. The report is structured and comprehensive. I have some remarks:
Methods: I would recommend to provide the composition of BEVOO.
Results: The changes in body composition could also be explained by by suboptimal measurement, as this technique is quite dependent on the total body water.
Discussion: do you have any hypothesis on the mode of action of BEVOO in this setting?
Reviewer 3 Report
Comments and Suggestions for Authors
The presented exploratory case study aims to investigate the use of BEVOO as an adjunct to nutritional intervention in a patient undergoing treatment for HPV 16–related tonsil cancer. By examining its potential complications and supporting recovery, this study seeks to contribute to the emerging field of personalized nutritional strategies in cancer care.
Some suggestions:
1.Methodology:
-You wrote: “The patient tracked the body composition and metabolic parameters daily at home using a multifunctional digital scale, the Vital Trainer Analytical Scale ETA 778090000, designed for precise weight measurement and body composition. Add please more details regarding the Vital Trainer Analytical Scale ETA 778090000
-line 128, you wrote “while weekly cisplatin (40 mg/m2) Monday to Friday”. What do you mean by Monday to Friday? Isn't 40 mg of cisplatin administered on the same day? Please clarify.
He was supposed to have 7 cycles of cisplatin administered once a week and he received only 2 cycles? Please clarify. - which is the origin of Bio ExtraVirgin Olive Oil consumed by the patient? Please add.
-line 137-138 you wrote “BEVOO was chosen for its high polyphenol content (over 250 mg/kg)”. Have you determined the amount of polyphenols in the consumed oil? Did you find this information on the product label? Please clarify.
- Please clearly specify how long the oil was used. You wrote “one week before therapy, and continuing throughout the treatment and recovery period”. What do you mean by recovery period?
2. In my opinion Table 3 and the discussions related to it must be presented at discussions. There are no results obtained by you. Are data taken from the literature.
- The composition of olive oil should be presented.
- Discussions:
Give please details concerning “Altogether, the patient subjectively reported that the consumption of BEVOO was the most tolerable nutritional component during treatment, which was supported by other investigations [28,29,30], causing minimal GI discomfort compared to other dietary interventions”
- Add please as Suppl Material the blank form of the questionnaires completed by the patient.
Round 2
Reviewer 1 Report
Comments and Suggestions for Authors
The authors have made appropriate revisions to the manuscript. However, I believe further modifications to the text are necessary.
The text is clear; however, in order to provide a more comprehensive overview, I recommend expanding lines 64–73 by incorporating recent studies on the effects of omega-3, particularly in the context of "Effect of omega-3 in patients undergoing bone marrow transplantation."
Methodology
Regarding my previous comment, I could not find any reference to the inclusion of a reporting guideline for your study in accordance with the EQUATOR Network.
References
Please review and align the references with the journal’s guidelines.
